# Analysis of Gravity Wave Characteristics during a Hailstone Event in the Cold Vortex of Northeast China

**Xiujuan Wang** [1,2], **Lingkun Ran** [3], **Yanbin Qi** [1,2,*], **Zhongbao Jiang** [4], **Tian Yun** [5] and **Baofeng Jiao** [3]

1   Jilin Province Technology Center for Meteorological Disaster Prevention, Changchun 130062, China
2   Joint Open Laboratory for Weather Modification of China Meteorological Administration,
    People's Government of Jilin Province, Changchun 130062, China
3   LACS, Institute of Atmospheric Physics, Chinese Academy of Sciences, Beijing 100029, China
4   Climate Center of Jilin Province, Changchun 130062, China
5   Meteorological Observatory of Jilin Province Meteorological Bureau, Changchun 130062, China
*   Correspondence: qiyanbin88@163.com; Tel.: +86-0431-8796-0078

**Abstract:** Based on high-resolution pressure data collected by a microbarograph and Fourier transform (FFT) data processing, a detailed analysis of the frequency spectra characteristics of gravity waves during a hailstone event in the cold vortex of Northeast China (NECV) on 9 September 2021 is presented. The results show that the deep NECV served as the large-scale circulation background for the hailstone event. The development of hailstones was closely related to gravity waves. In different hail stages, the frequency spectra characteristics of gravity waves were obviously different. One and a half hours before hailfall, there were gravity wave precursors with periods of 50–180 min and corresponding amplitudes ranging from 30 to 60 Pa. During hailfall, the center amplitudes of the gravity waves were approximately 50 Pa and 60 Pa, with the corresponding period ranges expanding to 60–70 min and 160–240 min. Simultaneously, hailstones initiated shorter periods (26–34 min) of gravity waves, with the amplitudes increasing to approximately 12–18 Pa. The relationship between hailstones and gravity waves was positive. After hailfall, gravity waves weakened and dissipated rapidly. As shown by the reconstructed gravity waves, key periods of gravity wave precursors ranged from 50–180 min, which preceded hailstones by several hours. When convection developed, there was thunderstorm high pressure and an outflow boundary. The airflow converged and diverged downstream, resulting in the formation of gravity waves and finally triggering hailfall. Gravity wave predecessors are significant for hail warnings and artificial hail suppression.

**Keywords:** gravity waves; microbarograph; the cold vortex of Northeast China; hailstone

## 1. Introduction

Hail events are a kind of natural disaster that frequently occur in the cold vortexes of Northeast China (NECV), with strong localization and rapid development. Large hail with diameters of greater than 2 cm can result in serious disasters. Many researchers, both domestic and international, have conducted extensive and in-depth studies on the radar characteristics of large hail [1,2], atmospheric stratification conditions of hailstones [3–5], and the three-dimensional structure of hailstones [6,7]. The formation mechanisms of hailstones are quite complex, so it is important to study the macrocharacteristics of hailstones, such as atmospheric formation conditions and echo characteristics. However, it is also necessary to further analyze the dynamic features of hailstones. Gravity waves are one of the major dynamic mechanisms that can trigger convection [8–10], but there have been few studies on gravity waves during hail processes [11–14]. Therefore, an in-depth study of gravity waves during NECV hail processes is scientifically significant for enhancing the forecasting of devastating hail events and disaster prevention and mitigation.

The chases and collisions of gravity waves are crucial mechanisms for the generation of convection, which also influence the propagation and distribution of convection. According

to Li [15] and Chao [16], when convection occurs, the ageostrophic balance can trigger gravity waves, which lead to latent heat release and convection generation. By simulating a mesoscale convective (MCS) event in the United States, Schumacher and Johnson [17] discovered that dynamic MCS conditions were insufficient with no cold pool near the ground, but that nearly static gravity waves in the lower layer organized and developed linear convection. According to a statistical analysis of 32 years of extreme rainstorm events in India [18], gravity waves were associated with updrafts in convective systems, which were the main causes of rainstorms. Ge et al. [19] statistically analyzed 12 local rainstorm events in Guangxi, finding that mesoscale eddies and gravity waves were the triggers for rainstorms. Liu et al. [20] simulated an orographic rainstorm in Sichuan, showing that the complex terrain caused heavy rainfall, accompanied by the generation of gravity waves. Apart from heavy rainfall events, gravity waves also play an important role in hailstone events. Using satellites, Putsay et al. [11] analyzed gravity waves during severe weather which included heavy precipitation, hailstone and strong winds. The results showed that gravity waves were clearly visible in MODIS imagery and were probably related to the pulsations of updrapt. Through hail simulation, Adamsselin [12] pointed out that the vertical motion of gravity waves was concentrated at lower levels. Taking advantage of microbarograph observations, Li et al. [13] and Ba et al. [14] found that there were gravity waves of large amplitudes during disastrous hailstone events. At present, the research objects of gravity waves are primarily rainstorms, while few works have focused on hailstones.

Due to economic and technological constraints, the methods for researching gravity waves primarily include theoretical analysis or mesoscale numerical simulation. Gravity waves have very broad frequency spectra, with periods ranging from several minutes to tens of hours. Gravity waves associated with mesoscale convection typically cover critical periods ranging from a few minutes to several hours [20,21]. Mesoscale numerical simulation with horizontal grid spacings from several kilometers to tens of kilometers can filter out key-scale gravity waves, so precise observations of gravity waves are quite important.

A microbarograph is a highly precise instrument used to detect gravity waves. The detection accuracy of a microbarograph can reach 1 Pa with a sampling frequency of approximately 0.1 Hz. As a result of their high sensitivity, microbarographs can clearly display gravity wave properties over broad periods ranging from minutes to hours [22]. A few researchers have utilized microbarographs to observe gravity waves. Using microbarograph and wind velocity fluctuation data, Gossard and Sweezy [23] investigated the gravity wave frequency spectra and wave dispersion. Grivet-Talocia and Einaudi [24] discussed in detail a technique of gravity wave identification based on microbarograph data, pointing out that the wavelet analysis method can effectively identify gravity waves. A microbarograph array with 500 m element spacing was used to statistically analyze gravity waves in the boundary layer [25]. According to the observations, gravity waves in the boundary layer have average periods of approximately 1–20 min and amplitudes of approximately 1–8 Pa. Li et al. [13] analyzed the frequency spectra of gravity waves of disastrous hailstone events using microbarograph data. They discovered that gravity waves appeared several hours before the hailstones fell and the amplitudes of the gravity waves were greater than 30 Pa. Based on a statistical analysis of the microbarograph data, the results revealed that gravity waves with amplitudes of greater than 30 Pa appeared 1–4 h earlier in two out of five hailstone events, with periods ranging from 30 to 70 min [14]. In NECV systems, the features of gravity waves during hailstone processes remain unclear.

Due to NECV, a large-scale hailstone event occurred in Jilin Province from 13:30 BJT (Beijing Time) to 15:00 BJT on 9 September 2021. The hailstones were mainly distributed across Changchun, Siping, Liaoyuan, and Jilin, which are the central regions of Jilin Province. Hailstone diameters generally ranged from 1–2 cm, and the maximum hailstone diameter even reached 5–6 cm. This hailstone event covered a broad area, which is relatively uncommon. This hailstone process was fully recorded by the microbarograph installed in Changchun. This paper uses high precision microbarograph data to analyze the

frequency spectral characteristics of gravity waves of the NECV hail process to provide a scientific basis for hail warnings. This paper is organized in the following manner: Section 2 introduces the microbarograph data from gravity wave observations and other meteorological data, and the methods used to analyze them; Section 3 provides the case study of the NECV hailstone event on 9 September 2021, Section 4 displays the characteristics and generation mechanism of gravity waves and summarizes the findings of this study; and Section 5 details the study conclusions.

## 2. Data and Method

A microbarograph (Figure 1) is a device used to observe low-frequency atmospheric disturbances on the ground, which can detect gravity waves from periods of several minutes to several hours. The microbarograph installed in Changchun was independently developed by the Key Laboratory of Cloud-Precipitation and Severe Storms (LACS), Institute of Atmospheric Physics, Chinese Academy of Sciences. The collected barometric data are highly accurate with a barometric resolution of approximately 0.1 Pa, a signal frequency of approximately $10^{-5}$ Hz and a sampling frequency of approximately 1 Hz. The pressure fluctuations caused by gravity waves are usually less than 100 Pa. In most cases, pressure fluctuations range from 1 to 10 Pa. Therefore, microbarographs can closely observe gravity wave characteristics.

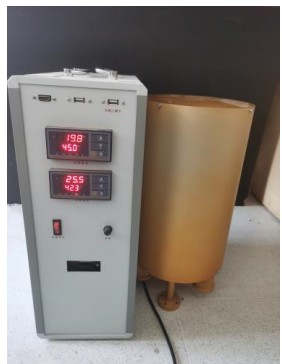

**Figure 1.** Microbarograph.

To obtain the spectral characteristics of gravity waves, the Fourier transform (FFT) method was applied to microbarograph data. The FFT is a signal analysis tool that reflects signals from the time domain to the frequency domain. Using the FFT method, microbarograph data were calculated every second to measure the wave amplitudes and periods up to the present time. Therefore, through visualizing the FFT results at each time step, the evolution of the gravity wave periods and amplitudes with time were obtained. The FFT defines $f(x)$ as a continuous function with only finite first-order discontinuity points and finite extreme points on any finite interval. If it is absolutely integrable on $(-\infty, +\infty)$, $f(x)$ is continuous at points as $f(x) = \frac{1}{2\pi} \int_{-\infty}^{+\infty} [\int_{-\infty}^{+\infty} f(t) e^{-i\omega t} dt] e^{i\omega x} d\omega$ and discontinuous as $f(x) = \frac{f(x+0) + f(x-0)}{2}$.

The FFT of $f(t)$ is defined as:

$$F(\omega) = \int_{-\infty}^{+\infty} f(t) e^{-i\omega t} dt \tag{1}$$

The inverse Fourier transform of the Fourier transform is defined as:

$$f(t) = \frac{1}{2\pi} \int_{-\infty}^{+\infty} F(\omega) e^{i\omega t} d\omega \tag{2}$$

The National Center for Environmental Prediction/National Center for Atmospheric Research (NCEP/NCAR) reanalysis data are used to calculate the water vapor flux of

the whole atmosphere layer, the isentropic potential vorticity, and the pseudoequivalent potential temperature.

The water vapor flux of the whole atmosphere layer represents the supply of water vapor. The zonal and latitudinal components of water vapor flux in the whole layer of atmosphere are shown, respectively, as follows:

$$Q_x = \frac{1}{g} \int_{p_s}^{p_t} qu\,dp \tag{3}$$

$$Q_y = \frac{1}{g} \int_{p_s}^{p_t} qv\,dp \tag{4}$$

where $p_t$ is the pressure of top layer, $p_s$ is the surface pressure, the zonal and latitudinal velocity are shown by $u$ and $v$, $g$ represent gravitational acceleration and $q$ is the specific humidity.

The spatial resolution of reanalysis data is $1° \times 1°$, with a time resolution of 6 h. The Atmospheric Administration GDAS assimilation data were used to analyze the mechanism of gravity waves, with the data spatial resolution of $0.25° \times 0.25°$ and a time resolution of 3 h.

### 3. NECV Hailstone Event

Influenced by the NECV system, there was a severe weather event in Jilin Province, including heavy rainfall, lightning, hailstones, a gale and other convective weather (Figure 2). Lightning occurred in most regions of Jilin Province, with rainstorms occurring at 18 stations, thunderstorms and gales appearing at 53 stations and short-term heavy rainfall occurring at 26 stations. The maximum 1 h accumulated rainfall occurred in Changchun, with values reaching 41 mm h$^{-1}$.

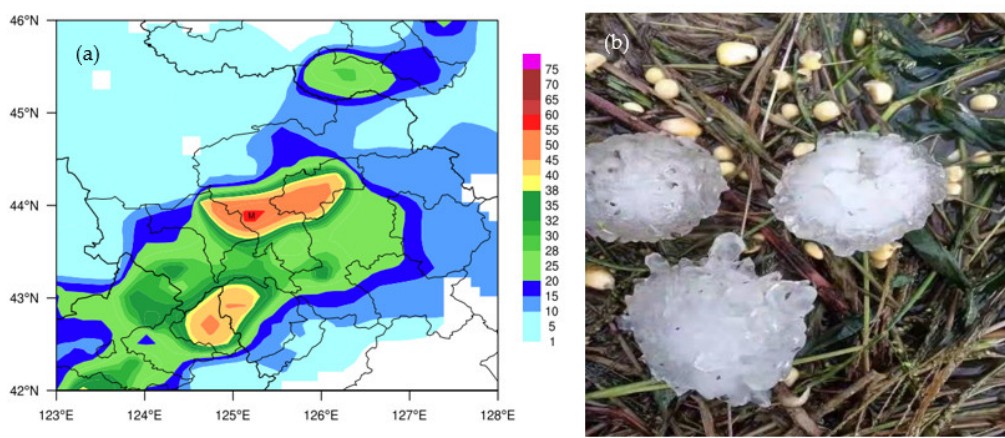

**Figure 2.** (**a**) Accumulated rainfall (shaded, mm) from 08:00 BJT 9 September to 08:00 BJT 10 September 2021, Changchun microbarograph (marked by M) and (**b**) hailstone.

Hail occurred in the central region of Jilin Province, in which the topography is very complicated (Figure 3). Severe weather often occurs in such region, including short intense precipitation, lightning, hail, downbursts and tornados. Overall, the terrain is flat in the northwest and high in the southeast (Figure 3). From 13:30 BJT to 15:00 BJT on 9 September, Changchun, Siping, Liaoyuan and Jilin were struck by hailstone events in succession. The hailstone diameters mostly ranged from approximately 1 to 2 cm, with maximum diameters reaching approximately 5–6 cm (Figures 2b and 3b). The heavy hailstone event was sudden and intense. Figure 3a shows the surrounding topography of the region of study. It is in the center of Northeast China in Jilin Province. Figure 3b shows that the terrain of hailstone regions has multi-scale features in Jilin Province; that is, it is flat in the west and on the

windward slope of Changbai Mountain in the east. The height difference in the terrain is significant.

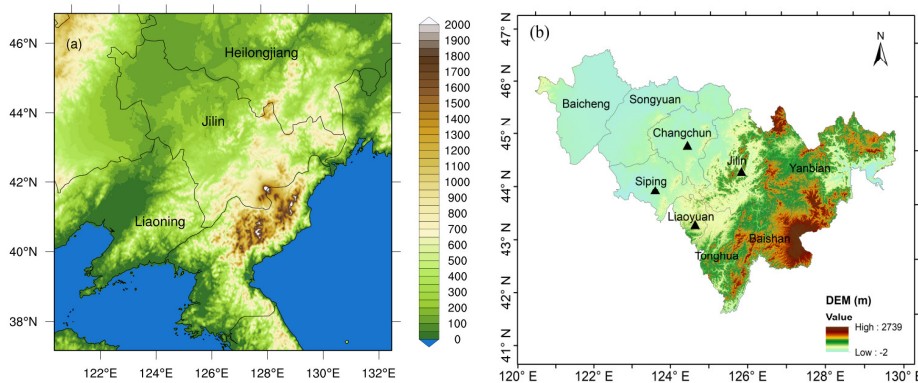

**Figure 3.** The topography of (**a**) the surrounding with terrain height in Jilin Province (m), (**b**) the terrain of the region of study in Jilin Province (black triangles denote the where hail occurred on 10 September 2021).

During the hailfall event, high-level weather conditions (Figure 4a,b) were at the front of the NECV and the back of the subtropical high in Jilin Province. The center of the NECV at 500 hPa coincided with that at the 850 hPa low level and they both occurred in the central regions of Inner Mongolia. The cores of the high-level and low-level cold vortices were in nearly the same area, indicating that the NECV system was relatively deep. Ridge Line 588 of the subtropical high showed an east–west pattern with the northern end of the ridge reaching close to 30° N. It was influenced by a southwest air flow in Jilin Province, which has an obvious dynamic lifting mechanism.

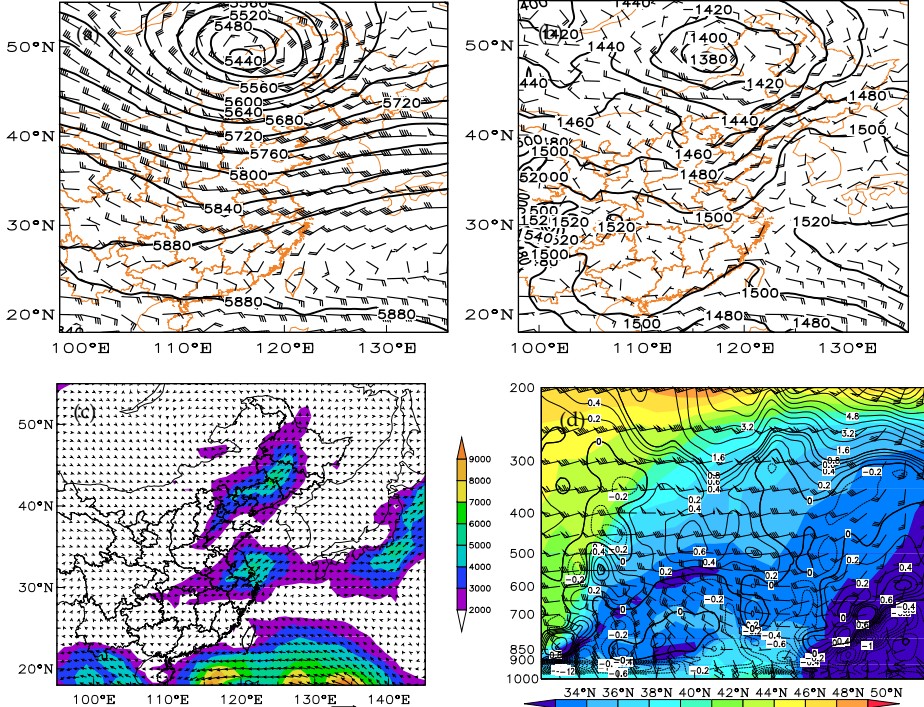

**Figure 4.** (**a**) 500 hPa geopotential height field (black lines, gpm) and wind field (wind stick, m/s) at 08:00 BJT; (**b**) 850 hPa geopotential height field (black lines, gpm) and wind field (wind stick, m/s) at 08:00 BJT; (**c**) whole-layer atmospheric moisture flux at 14:00 BJT; (**d**) vertical section of the isentropic potential vorticity, pseudoequivalent potential temperature and wind field along 125° E at 14:00 BJT on 9 September 2021.

Water vapor was transported to Jilin Province by the southwest airflow behind the subtropical high. The water vapor flux of the whole atmosphere layer was calculated from the ground to 100 hPa (Figure 4c). On 14:00 BJT 9 September, the maximum value of the whole layer water vapor flux occurred in the central parts of Jilin Province, namely Changchun, Siping, Liaoyuan and Jilin, with values of approximately 6000 g·(cm·s·hPa)$^{-1}$.

To identify unstable conditions, the isentropic potential vorticity, the pseudoequivalent potential temperature, and the wind field vertical profile (Figure 4d) were measured along 125° E in the hailstone region. When hailstones fell at 14:00 BJT on 9 September, unstable atmospheric stratification ($\frac{\partial \theta_{se}}{\partial p} > 0$) appeared in the lower layer and stable atmospheric stratification ($\frac{\partial \theta_{se}}{\partial p} < 0$) was distributed in the middle and upper layers. The brown solid line in Figure 4d represents the front zone of the pseudoequivalent potential temperature, in which the pseudoequivalent potential temperature decreases with height. The atmospheric stratification is unstable below the front zone of $\theta_{se}$. The front zone of $\theta_{se}$ was at approximately 850 hPa (Figure 4d) and sloped to the south with increasing elevation, indicating that the low atmospheric stratification was unstable. In the hailstone zone, the middle and upper levels (above 500 hPa) were positive potential vorticity areas, where westerly winds prevailed and cold, dry air with an intensity of 0 PVU invaded along the frontal region. In general, the lower layer showed unstable atmospheric stratification along with the intrusion of dry and cold air, which generated unstable energy.

Convective cloud A, which was in the north–south direction, echoes approximately 60 dBz with a cloud height of about 14 km, and was formed in eastern Changchun from 10:03 BJT on 9 September (Figure 5a). Convective cloud A reached Changchun at 10:03 BJT and continued to proceed easterly at 11:28 BJT. From then on, convective cloud A began to weaken in Changchun. By 12:10 BJT, convective cloud A had fully moved out of Changchun. There was no radar echo in Changchun until 12:43 BJT. Convective cloud B (Figure 5b) formed in Changchun at approximately 12:43 BJT due to favorable thermal conditions. The echo intensity of convective cloud B was as high as 60 dBz with a scale smaller than that of convective cloud A. The height of convective cloud B reduced to 10 km (Figure 5f). Convective cloud B moved eastward and left Changchun at 14:03 BJT. Convective cloud C (Figure 5c) developed in Changchun at the same time with large amounts of convective cells forming around Changchun. Convective cloud C moved eastward and merged with the surrounding strong convection after 14:00 BJT, finally forming convective cloud D at 15:02 BJT (Figure 5d). Hailstones fell in Changchun, Siping, Liaoyuan and Jilin from 13:30 BJT to 15:00 BJT as a result of convective cloud B,C and D.

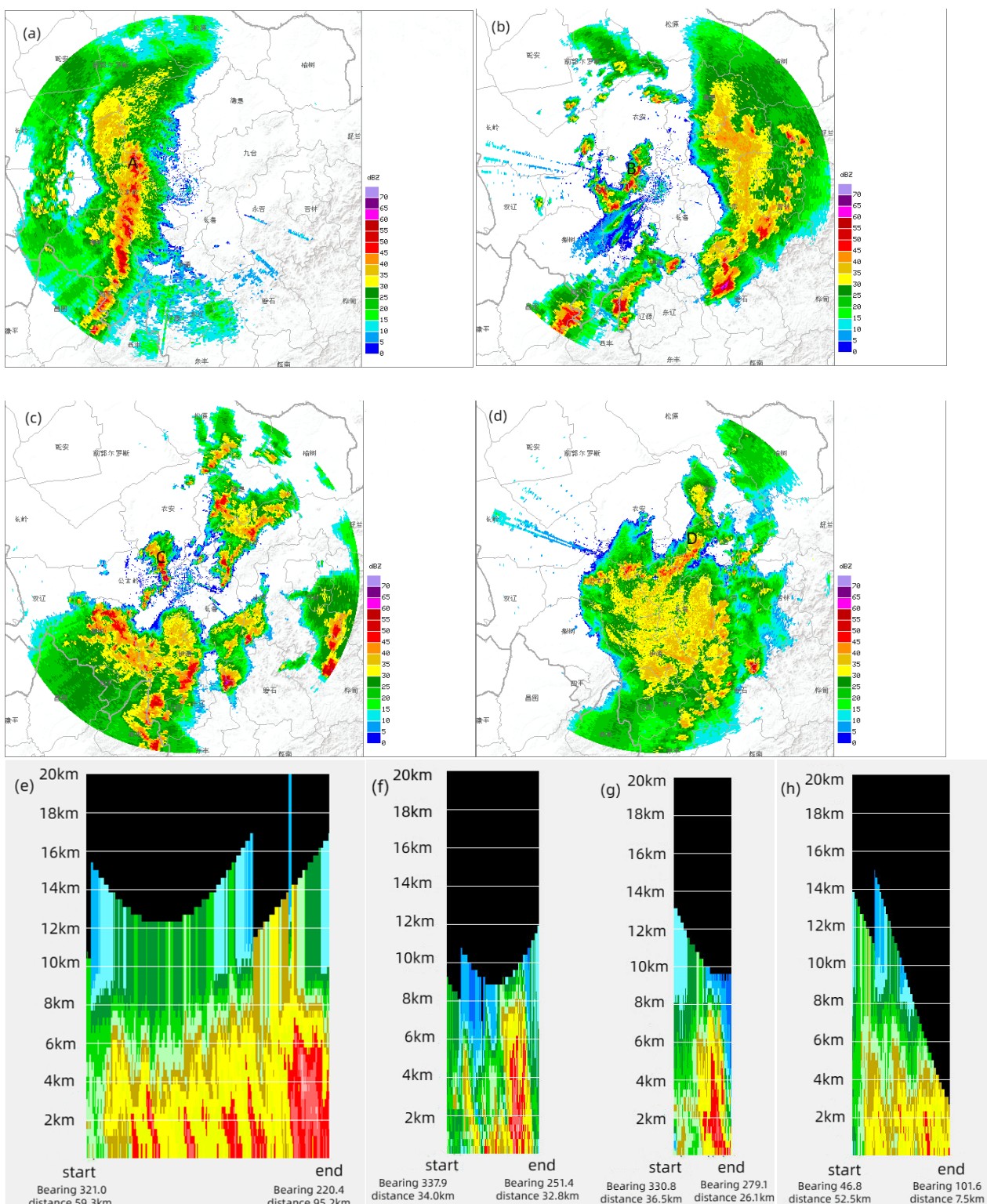

**Figure 5.** Radar reflectivity and vertical profile along convective cloud A, B, C and D, respectively, at (**a**,**e**) 10:03 BJT; (**b**,**f**) 12:43 BJT; (**c**,**g**) 14:03 BJT and (**d**,**h**) 15:02 BJT on 9 September 2021. In figure (**e**–**h**), the units of bearing is degree.

## 4. The Characteristics and Generation Mechanism of Gravity Waves

### 4.1. Gravity Wave Characteristics in the Time Domain

Minute-scale pressure disturbances that occur during a convection process may represent the variable properties of convective systems in detail. Figure 6a depicts the time series variation and daily variation in pressure obtained from the microbarograph. Moving average filtering was applied to microbarograph data to calculate the daily pressure change with the data period of 08:00 BJT on 8 September to 08:00 BJT on 10 September and a window width of 18,000. A moving average filter is a low-pass filter. Assume the input data are denoted by $x$, the output data are denoted by $y$ and the window width is $N$. First, the initial data is removed from the last window queue for a new calculation. Second, the remaining data $N - 1$ are advanced forward in turn. Third, the new calculation data are inserted as the tail of the new queue. The moving average filter calculation formula is:

$$y(n) = \frac{x(n) + x(n-1) + x(n-2) + \ldots + x(n-N-1)}{N} \tag{5}$$

**Figure 6.** Microbarograph data: (**a**) pressure time series (black line, hPa), daily pressure variation (red line, hPa), (**b**) disturbance pressure (black line, Pa) and hourly accumulation rainfall (green line, mm). The black triangle denotes the onset of hail at Changchun station on 8–10 September 2021.

Figure 6b depicts the disturbance pressure produced by subtracting the daily pressure variation from the pressure time series with an overlay of the hourly accumulation rainfall.

Figure 6 shows that pressure rose from 18:00 BJT on 8 September to 06:00 BJT on 9 September with multiscale pressure disturbances. When there was no accumulated rainfall (00:00 BJT–10:00 BJT 9 September), the pressure disturbance ranged from approximately −40 to 40 Pa. The pressure presented a substantial decreasing trend from 06:00 BJT to 13:00 BJT on the 9th, thus just before the hailstone event. The pressure disturbance was intense from 10:00 BJT to 13:00 BJT on the 9th. In particular, the pressure jumped dramatically to 985.5 Pa at 10:00 BJT and then dropped sharply to 981 Pa at 11:00 BJT. There was a small amount of rainfall at this time, when the disturbance pressure decreased abruptly from 250 Pa to −100 Pa. During hailstone occurrence, the pressure values stabilized at 980.5–982.5 Pa from 13:00 BJT–16:00 BJT on 9 September with disturbance pressure levels of approximately −70–70 Pa. After hail had fallen, the pressure showed an increasing trend from 16:00 BJT on 9 September to 06:00 BJT on 10 September, with the disturbed pressure gradually decreasing to −20–20 Pa. Pressure levels fluctuated significantly before and during the hailstone event. Two-and-a-half hours before hail occurred (i.e., 11:00 BJT on the 9th), there was a sharp fluctuation in pressure. Such an intense variation in pressure carries a specific precursory meaning for hail.

### 4.2. Gravity Wave Characteristics in the Frequency Domain

The time-domain characteristics of waves show that there were gravity waves during the hailstone event. To analyze the frequency spectral characteristics of waves, pressure data collected by the microbarograph were transformed using the FFT (Figure 7). Figure 7b

focuses on the frequency spectral characteristics of gravity waves with periods of less than 80 min to analyze the frequency spectral characteristics in more detail.

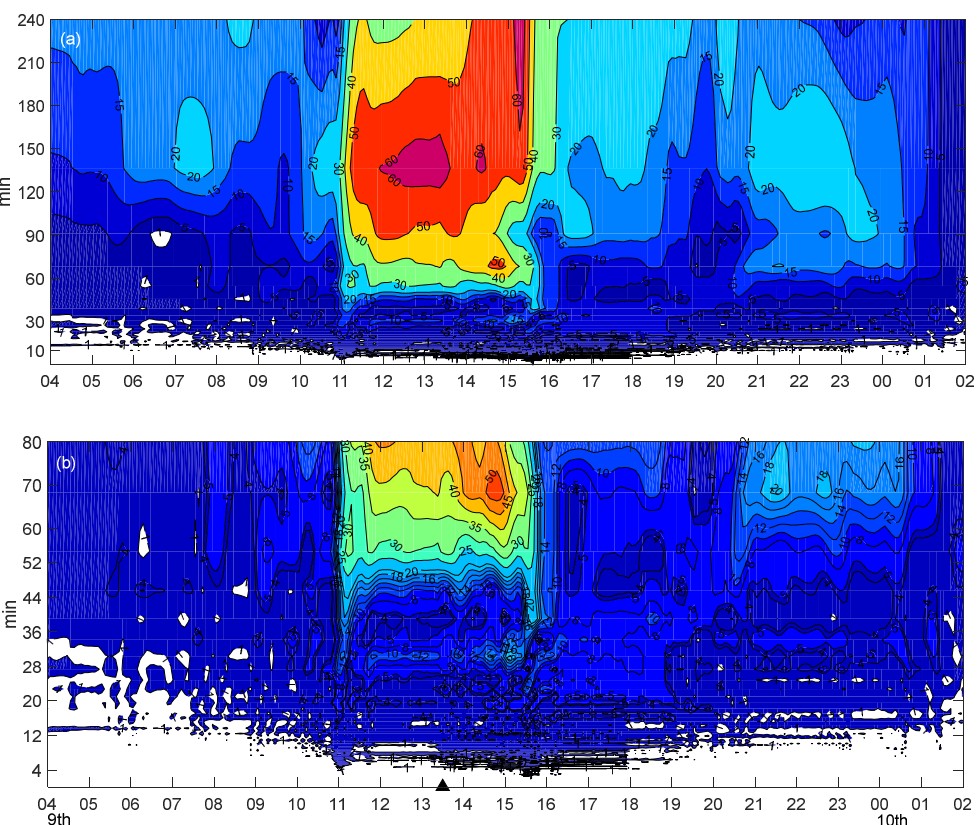

**Figure 7.** The gravity wave frequency spectra on 9–10 September 2021: (**a**) periods shorter than 240 min and (**b**) periods shorter than 80 min. In the figure, the x-axis denotes time (units: hour), the shading denotes the amplitudes of gravity waves (units: Pa), the y-axis denotes the periods of gravity waves (units: min), and the black triangle denotes the onset time of hailfall.

According to Figure 7a, the amplitudes of gravity waves with periods of approximately 1–240 min were 1–60 Pa. Gravity waves were evolved primarily in three stages: 04:00–11:00 BJT 9 September, 11:00–16:00 BJT 9 September and from 16:00 BJT 9 September to 00:00 BJT 10 September. At each stage, the amplitude of the gravity wave increased and then decreased. To further investigate the relationship between hail and gravity waves, the evolution of gravity waves was divided into three stages based on the time of occurrence of hail.

Before the hailstone event (04:00 BJT to 13:30 BJT 9 September), the wave amplitudes began to increase at 05:30 BJT and reached 20 Pa from 07:00 to 08:00 BJT. There was no echo in Changchun at this time, and scattered echoes formed and quickly dissipated approximately 60 km southwest of Changchun (sketch). Periods of 220–240 min of gravity waves with amplitudes of approximately 20 Pa appeared from 08:00 BJT to 09:00 BJT. There was still no echo in Changchun at this time, but convective cloud A was moving eastward at approximately 50 km southwest of the city (sketch). Then, the amplitudes of the waves began to decrease, with values of less than 20 Pa. Convective cloud A moved to the east of Changchun at 10:03 BJT (Figure 5a), at which time the amplitudes of the gravity waves increased to more than 20 Pa, corresponding to periods of approximately 120–160 min. Convective cloud A continued to move eastward at 11:00 BJT and gravity waves with periods of 1–240 min quickly developed. From 11:00 to 12:00 BJT, the amplitudes of the gravity waves with periods of 80–180 min were the largest, with values of approximately 50–60 Pa. In particular, the amplitudes of the gravity waves with periods of approximately 120–160 min formed a closed center, reaching 60 Pa. At this time, the amplitudes of the

gravity waves with shorter periods of 50–80 min increased significantly to approximately 30–50 Pa. Convective cloud A moved eastward and weakened continuously from 11:00 to 12:00 BJT (sketch). At this time, the maximum radar reflectivity of convective cloud A was approximately 60 dBz in Changchun. By 12:10 BJT, convective cloud A had completely moved out of Changchun, and there was no echo. Simultaneously, the gravity waves continued to strengthen in each period. The closed center amplitudes of the gravity waves ranged from approximately 50 to 60 Pa, and the corresponding periods were still approximately 80–180 min. The amplitudes of gravity waves with periods of approximately 50–80 min also remained, with values of approximately 30–50 Pa. Simultaneously, shorter periodic gravity waves were excited, resulting in gravity waves with periods of approximately 26–32 min and amplitudes of approximately 10–12 Pa. At 12:43 BJT, convective cloud B formed in Changchun (Figure 5b) and hailstones fell at 13:30 BJT.

Gravity waves appeared before convection, which is consistent with previous research results [26,27]. Compared to previous studies, the differences are that the periods of gravity wave precursors before the rainstorm were approximately 140–270 min, whereas the periods of gravity wave precursors before the hailstone event were shorter, concentrating at 50–180 min.

During the hailstone event (13:30 BJT to 15:00 BJT 9 September), gravity waves continued to develop. In contrast to patterns observed before the hailstone event, the closed center amplitudes of gravity waves with periods of 120–160 min gradually disappeared. The center amplitudes of the gravity waves were concentrated at 160–240 min and 60–70 min, with values of approximately 60 Pa and 50 Pa, respectively. Another remarkable feature is that the larger amplitudes of gravity waves in shorter periods (26–34 min) were instigated, with amplitudes increasing to 12–18 Pa. During the hailstone event, convective clouds B, C and D successively appeared in Changchun (Figure 5) with echo intensities reaching up to 60 dBz. As observed, the formation of convective clouds promoted the development of gravity waves.

After the hailstone event (15:00 BJT 9 September to 00:00 BJT 10 September), convective cloud D weakened and moved eastward at 15:02 BJT (Figure 5d), completely moving out of Changchun at 16:30 BJT. The gravity waves likewise diminished rapidly with central amplitudes reaching less than 20 Pa.

According to the above analysis, there was gravity wave activity 1.5 h before the hailstone event, with corresponding periods of approximately 50–180 min and center amplitudes of approximately 30–60 Pa. During hailfall, gravity waves also developed. There were center amplitudes of gravity waves of approximately 50 Pa and 60 Pa with corresponding periods of 60–70 min and 160–240 min. Simultaneously, the amplitudes of gravity waves of approximately 12–18 Pa were instigated, with periods ranging from 26 min to 34 min. The precursor signals of gravity waves occurred approximately 1.5 h ahead of the hailstone event, indicating that the precursor activities of gravity waves preceded the hailstone event and played a positive role in triggering hailfall. This is in agreement with previous studies [13,14].

*4.3. The Reconstruction of Gravity Wave Precursors*

Figure 7 shows that the wave precursor signals with periods of approximately 50–180 min were the most obvious. Studying the properties of the precursor signals involved two steps. First, gravity waves in this frequency range (50–180 min) were extracted. Second, the precursor signals of gravity waves were reconstructed by the inverse Fourier transform method. The reconstructed gravity waves (Figure 8) exhibited periodic disturbances during 00:00–09:00 BJT 9 Sep with disturbance amplitudes of less than 10 Pa. During 09:00–10:00 BJT 9 Sep, the amplitudes began to quickly grow, with extreme values of approximately 30 Pa. There was a small amount of accumulated rainfall between 10:00 and 13:00 BJT, while the wave precursor signals were obvious at this time. From 10:00 to 11:00 BJT, the disturbance amplitudes first decreased to −50 Pa and then rapidly increased to 128 Pa. From 11:00 to 12:00 BJT, the wave amplitudes abruptly decreased to −82 Pa. The amplitude

ranges were stable at −10–10 Pa from 12:00 to 13:00 BJT. When hailfall occurred, the disturbance amplitudes increased significantly from 13:00 BJT to 15:00 BJT, with values ranging from −60 to 65 Pa. Under the influence of weakened convection, the fluctuation amplitudes ranged from −30 to 30 Pa from 15:00 to 20:00 BJT. After 20:00 BJT, convection dissipated, and the fluctuation amplitudes were basically maintained at −10–10 Pa.

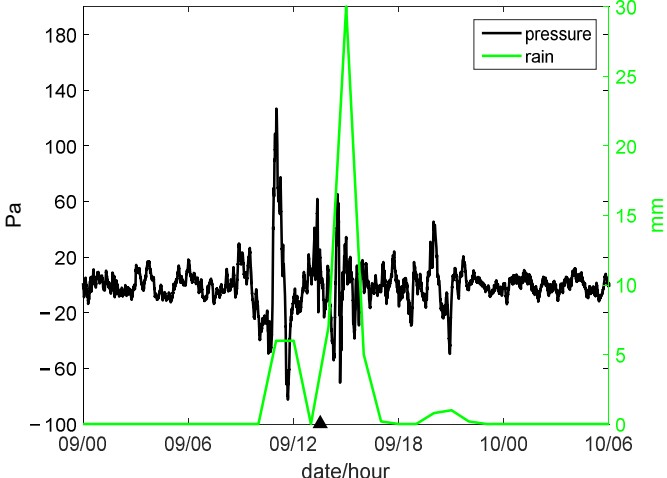

**Figure 8.** The reconstruction of gravity wave precursors (black line, Pa) and hourly accumulation rainfall (green line, mm) from 00:00 BJT on 9 September to 06:00 BJT on 10 September. The black triangle denotes the onset of hailfall.

According to the above analysis, the precursor signals of reconstructed waves with periods of 50–180 min appeared before hailfall. The reconstructed waves fluctuated sharply, increasing significantly from −50 Pa to 128 Pa and then dropping considerably to −82 Pa from 10:00 BJT to 12:00 BJT on 9 September. Such precursor signals of reconstructed waves appeared approximately 2.5 h ahead of the hailstone event. During the hailstone event, the reconstructed waves fluctuated greatly from −60 to 65 Pa. The gravity wave precursors are predictive of hail.

*4.4. The Mechanism of Generation of Gravity Waves*

In the discontinuous layer of the stratified stable atmosphere, the air micromass deviates from its equilibrium position, then gravity waves can be generated by the influence of gravity and buoyancy. The gravity wave equations [15] are as follows:

$$\begin{aligned}
\frac{\partial u'}{\partial t} - fv' &= -\frac{\partial p'}{\partial x}, \\
\frac{\partial v'}{\partial t} + fu' &= -\frac{\partial p'}{\partial y}, \\
\frac{\partial w'}{\partial t} - g\theta' &= -\frac{\partial p'}{\partial z}, \\
\frac{\partial u'}{\partial x} + \frac{\partial v'}{\partial y} + \frac{\partial w'}{\partial z} &= 0,
\end{aligned} \tag{6}$$

In the above formulas, $u'$, $v'$ and $w'$ represent the disturbance in wind speed in the direction of $x$, $y$ and $z$. $f$ is the Coriolis parameter. $\theta'$ is the potential temperature disturbance. $K_x$, $K_y$ and $K_z$ are the wave number in the direction of $x$, $y$ and $z$, respectively. The formula of frequency is:

$$\omega^2 = \frac{K_z^2 f^2 + (K_x^2 + K_y^2)N^2}{K_x^2 + K_y^2 + K_z^2} \tag{7}$$

The formulas for divergence, vorticity and pressure disturbance can be derived from the above formulas:

$$
\begin{aligned}
D &= -K_z W_0 \cos K_z z \sin(\omega t - K_x x - K_y y), \\
\zeta &= -\frac{W_0 f K_z}{\omega} \cos K_z z \cos(\omega t - K_x x - K_y y), \\
p' &= -\omega\left(1 - \frac{f^2}{\omega^2}\right)(K_x^2 + K_y^2)^{-1} K_z W_0 \cos K_z z \cos(\omega t - K_x x - K_y y).
\end{aligned}
\tag{8}
$$

As seen in Formulas (6) and (8), the pressure disturbance can reflect gravity wave features. Horizontal convergence, divergence and vorticity are closely related to the formation of gravity waves. Horizontal convergence and divergence can cause ascending and descending motion, finally triggering gravity waves under the effect of gravity. The mechanism of the generation of gravity waves is discussed below by using surface pressure disturbance, surface wind field, divergence disturbance and temperature disturbance (Figure 9).

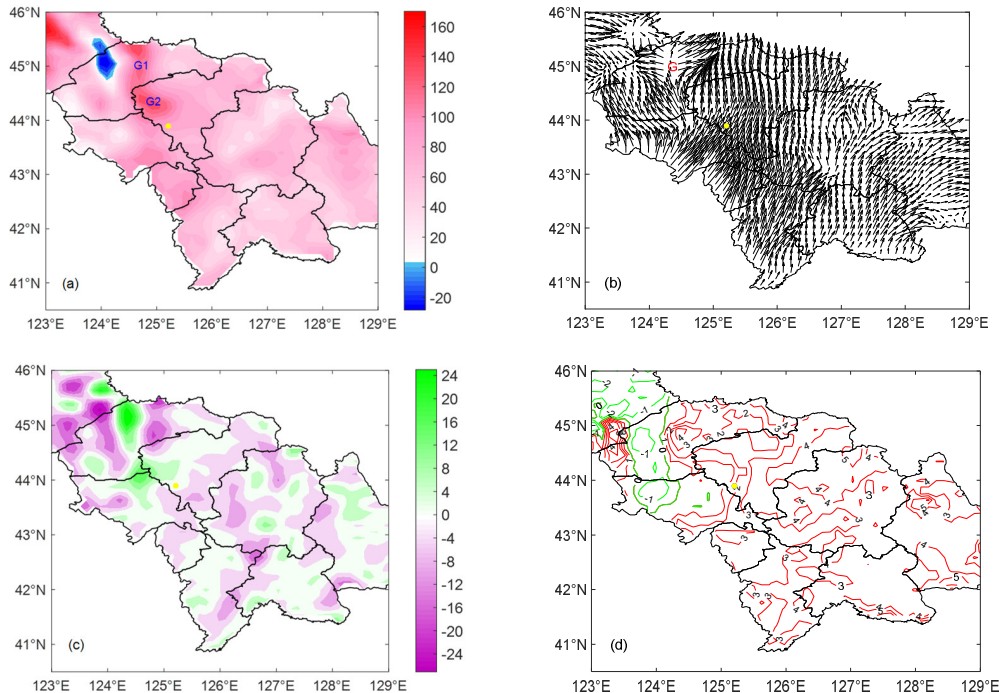

**Figure 9.** (**a**) Surface pressure disturbance (Pa), (**b**) surface wind field, (**c**) divergence disturbance and (**d**) temperature disturbance (green solid lines denote positive values and red dotted lines denote negative values, units: °C) at 11:00 BJT on 9 September 2021. The yellow dot denotes the location of the microbarograph.

The strong echo was mainly distributed in Songyuan, Changchun and Siping, which are in the central and western regions of Jilin Province, at 11:00 BJT on 9 September (Figure omitted). Figure 9a shows that there were high pressures, G1 and G2, as a result of precipitation particle sinking movement. There was airflow divergence around G1 with the direction of wind from west, east and south (Figure 9b). The airflow converged and diverged in the downstream of convection (Figure 9c), which was across Changchun and Siping. Cold air invaded at the rear of G1 (Figure 9d). The regions of convergence and divergence present the shape of band and propagated to the southeast. Such a phenomenon reflects the characteristics of gravity waves.

The downstream outflow was primarily produced by the surface high pressure caused by downdraft and precipitation evaporation. In accordance with the mass continuity principle, the flow of downstream diverged. Therefore, the band distribution of divergence

and convergence spread outward. Finally, gravity waves formed. Given that gravity waves formed in downstream of convection, they propagated before convection, finally exerting a positive effect on the generation of hailstones. Such a feature of gravity waves may be a precursory indicator for hail.

### 4.5. The Comparision of Gravity Wave Characteristics with Those of Heavy Rainfall and Hailstone Events

In order to better understand the differences in gravity wave characteristics between those of heavy rainfall and hail, a heavy rainfall event which occurred on 30–31 July 2021 in Changchun is displayed below. The hourly accumulation rainfall (Figure 10b) showed that the heavy rainfall mainly occurred from 14:00 BJT 30 July to 06:00 BJT 31 July, with the maximum hourly accumulated rainfall reaching 14 mm h$^{-1}$ at 23:00 BJT on 30 July.

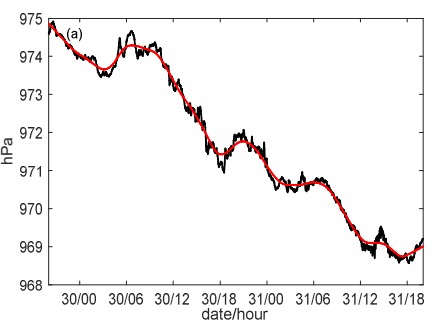 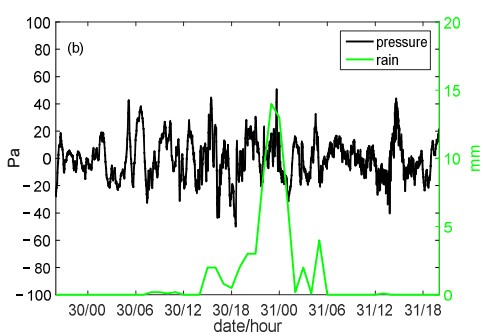

**Figure 10.** Microbarograph data: (**a**) pressure time series (black line, hPa), daily pressure variation (red line, hPa), (**b**) disturbance pressure (black line, Pa), hourly accumulation rainfall (green line, mm) at Changchun station on 30–31 July 2021.

The microbarograph which is installed in Changchun fully documented this event. Moving average filtering was applied to calculate the microbarograph data. Therefore, the pressure time series, daily pressure variation and disturbance pressure were obtained (Figure 10). The daily pressure variation presents a decreasing trend from 20:00 BJT 29 July to 20:00 BJT 31 July, including small multi-scale disturbances (Figure 10a). The hourly accumulated rainfall (Figure 10b) shows that there was one rainfall process, with the peak rainfall happening from 22:00 BJT 30 July to 00:00 BJT 31 July. Figure 10b shows that the disturbance pressure appeared in advance of the peak rainfall. The hourly accumulation rainfall reached a peak at 23:00 BJT, but the disturbance pressure developed from 05:00 BJT 30 July with a value of about 43 Pa. The pressure fluctuated periodically from then on, with the ranges of disturbance amplitudes from 44 Pa to −50 Pa until the occurrence of peak rainfall. The precursory feature of gravity waves is clear for this heavy rainfall.

The method of FFT was applied to microbarograph data to obtain the frequency spectral characteristics of gravity waves (Figure 11). Before the heavy rainfall (14:00 BJT 30–06:00 BJT 31 July), there were periodic disturbances in gravity waves with center amplitudes of approximately 10–16 Pa and periods ranging from 80 to 240 min during 05:00 BJT–12:00 BJT 30 July. During the heavy rainfall, the gravity waves developed, with amplitudes of about 10–16 Pa and the corresponding ranges of periods from 80 to 240 min.

Compared with the characteristics of gravity waves in hailstone events, the features of this heavy rainfall case present differently in two aspects. Firstly, the amplitudes of the gravity waves in the hailstone event were obviously larger than those in heavy rainfall. Secondly, the periods of the gravity waves in the hailstone event were shorter than those in the rainstorm event. Gravity wave precursors were the common feature both in heavy rainfall and hailstone events. This is predictive of heavy rainfall and hail events.

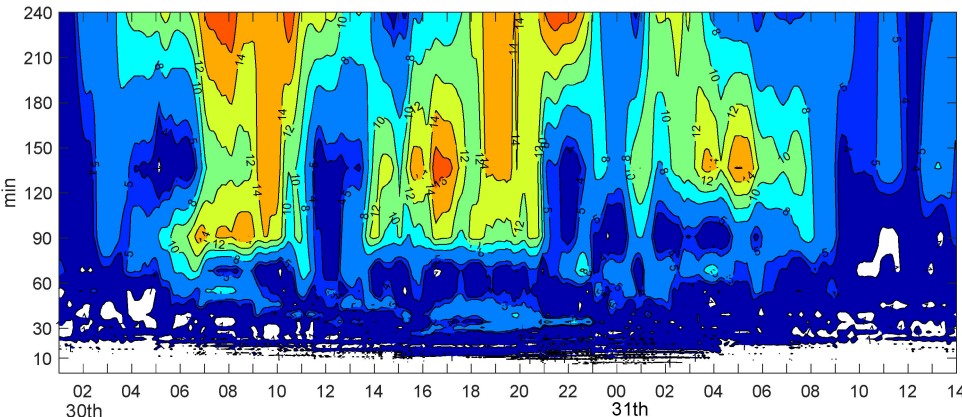

**Figure 11.** The gravity wave frequency spectra on 30–31 July 2021: periods shorter than 240 min. The figure illustration is the same as Figure 7.

## 5. Conclusions

The characteristics of gravity waves during a hailstone event in the NECV on 9 September 2021 are analyzed, for which microbarograph data independently developed by the LACS and observation data from the Automatic Ground Station of Jilin Province are both used. The following conclusions are summarized:

(1) The deep NECV served as the large-scale circulation setting for the hailstone event. The water vapor flux of the whole atmospheric layer demonstrates that the southwest airflow surrounding the subtropical high transported a large amount of water vapor for this convection. During the hailstone event, a positive potential vorticity was disturbed in the top layer and slid along the pseudopotential temperature front. The stratification of the atmosphere was unstable. The hailstone event occurred suddenly and with high intensity.

(2) The time-domain and frequency-domain properties of gravity waves revealed a strong relationship between hailstone and pressure disturbances. A period of 1.5 h before the occurrence of hail, there were gravity wave precursor activities with periods of approximately 50–180 min and amplitudes of approximately 30–60 Pa.

(3) The gravity wave frequency spectrum suggests that hailstones can enhance the development of gravity waves. During hailfall, there were gravity waves with periods of 60–70 min and 160–240 min, with corresponding amplitudes of approximately 50 Pa and 60 Pa, respectively. Gravity waves with shorter periods of 26–34 min were also triggered, with amplitudes of 12–18 Pa. After hailfall, gravity waves weakened. The relationship between hailstone fall and gravity waves was positive.

(4) The reconstructed precursors of gravity waves show that the key periods ranged from 50–180 min, accompanied by violent pressure fluctuations that occurred 2.5 h ahead of the hailstone event. Gravity waves preceded the hailstone event by several hours, which is predictive of hail events.

(5) During the convection development, precipitation particle sinking and precipitation evaporation generated high pressure near the ground, with the outflow diverging outward. This resulted in the convergence of downstream airflow. According to the principle of mass continuity, the flow ascends in the convergence zone and descends in divergence zone. The flow of convergence and divergence generates gravity waves as well as downstream convection. This was the primary formation mechanism of gravity waves, which triggered the downstream convection and resulted in hail.

(6) A heavy rainfall event that occurred during 30–31 July 2021 is presented to compare the features of gravity waves with those of the hailstone event. The results showed that there were gravity wave precursors in both cases, and the amplitudes of the gravity waves in the hailstone event were larger than those in the heavy rainfall event and the periods of gravity waves were shorter.

**Author Contributions:** Conceptualization, X.W. and L.R.; methodology, L.R.; validation, Z.J. and T.Y.; formal analysis, X.W. and B.J.; resources, X.W.; data curation, X.W.; writing—original draft preparation, X.W.; writing—review and editing, L.R. and Y.Q.; visualization, Y.Q.; supervision, L.R. All authors have read and agreed to the published version of the manuscript.

**Funding:** This research was funded by the Strategic Priority Research Program of the Chinese Academy of Sciences, grant number XDA17010105, the National Natural Science Foundation of China, grant number 41775140, Scientific and Technological Developing Scheme of Jilin Province, grant number 20230203126SF, and the Scientific Research Project of Jilin Province Meteorological Bureau, grant number 202109.

**Institutional Review Board Statement:** Not applicable.

**Informed Consent Statement:** Not applicable.

**Data Availability Statement:** Not applicable.

**Acknowledgments:** The authors would like to thank all those who contributed to this research.

**Conflicts of Interest:** The authors declare no conflict of interest.

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
