# Peer review of "Analysis of Gravity Wave Characteristics during a Hailstone Event in the Cold Vortex of Northeast China"

_atmosphere, doi:10.3390/atmos14020412_

Round 1

Reviewer 1 Report

Dear Authors,

Please find attached my comments

Best regards

Author Response

Response to Reviewer 1 Comments

Dear reviewer,

Thank you very much for your precious comments. According to your opinions, we have revised manuscript carefully. Hope it more suitable for publication. The revised part of article is highlighted in blue font. Moreover, due to the addition of references and figures, the serial number of references and figures in the most text have also modified.

Reviewer 1

  • Point 1: Abstract: You should include some words regarding the methodology: how the Authors have detected or estimated the gravity waves?

Reponse: We have added the methodology in L14-15. 

  • Point 2: Introduction: The Authors say L38 “… to further analyze the dynamic features of hailstones. Gravity waves are one of the major dynamic mechanisms that can trigger convection [8–10], but there have been few studies of gravity waves during hail processes.” The Authors need to link convection with hailstorms because the references are linked to rain episodes. Besides, which are the studies of gravity waves in hail events? (I can’t find them in the bibliography). Similar issue in L54.

Reponse: We have added the references of hailstorms, with the serial numbers from 11 to 14. Related revision is in L46. The details of hailstone references are depicted in L64-70, L93-99.

  • Point 3: I suggest split in two the paragraph L56-81: L63 divides the paragraph.

Reponse:We have split it into two paragraphs.

  • Point 4: Please, provide a general map of the surrounding to the region of study (maybe of the China and the neighborhoods) and another more specific of the region of study, with the complete toponomy included in the manuscript.

Reponse: We have added the toponomy in Figure 3, including the surrounding with terrain (Figure 3(a)) and the region of this study with terrain (Figure 3(b)). Relevant description has been added in L172-181.

  • Point 5: Figure 2: “Marked by Mic”?

Reponse: We have replaced symbol “Mic” with “M” in Figure 3 (original number of figure is 2). Symbol “M” represents microbarograph.

  • Point 6: L148: “Measured” or “Estimated”?

Reponse: The whole layer water vapor flux is calculated, we have added the calculation formulas in L146-153.

  • Point 7: Figure 3: what is the brown line in panel (d)?

Reponse: The explanation of the brown line in figure 4d (original number of figure is 3d) has been added in L205-208. The details are as follows: The brown solid line in Figure 4d represents the front zone of pseudoequivalent potential temperature, in which the pseudoequivalent potential temperature decreases with height. The atmospheric stratification is unstable below the front zone of .

  • Point 8: Figure 4 and adjacent text: I think that B, C and D cannot be considered “squall lines”. Which are the dimensions of the three structures?

Reponse: Convective cloud may be more accurate. We have replace “squall lines” with “convective cloud”. The vertical section of radar echo has been added in figure 5(e,f,g,h) to better display the dimensions of the three structures. Relevant description is in L223-231.

  • Point 9: Figure 5: Could you move the black triangle a bit out of the label?

Reponse: We have corrected the location of the black triangle.

Reviewer 2 Report

Overall, the manuscript is very well written, very interesting, and reports new results on the Analysis of Gravity Wave Characteristics During a Hailstone  Event in the Cold Vortex of Northeast China.  The authors put a lot of work and effort into obtaining the high precision microbarograph data to analyze the frequency spectrum characteristics of gravity waves of the NECV hail process to provide a scientific basis for hail warning. The results presented are high quality and the manner of presentation is very professional and precise.  The manuscript is certainly recommended for publication. 

There are minor corrections recommended that the authors can complete during the process of correction/proofreading. It would be good if authors write a sentence at the end of the Introduction that describes the structure of the manuscript. Also references should be expend with ones which describe a similar problem in a different way and current state in this area. Please expend and better explain the method i.e. the section with FFT. Figures 5 and 7 need title of X-axis. It would also be a good to read the text carefully  to correct English and typos.

Author Response

Response to Reviewer 2 Comments

Dear reviewer,

Thank you very much for your precious comments. According to your opinions, we have revised manuscript carefully. Hope it more suitable for publication. The revised part of article is highlighted in blue font. Moreover, due to the addition of references and figures, the serial number of references and figures in the most text have also modified.

Overall, the manuscript is very well written, very interesting, and reports new results on the Analysis of Gravity Wave Characteristics During a Hailstone  Event in the Cold Vortex of Northeast China.  The authors put a lot of work and effort into obtaining the high precision microbarograph data to analyze the frequency spectrum characteristics of gravity waves of the NECV hail process to provide a scientific basis for hail warning. The results presented are high quality and the manner of presentation is very professional and precise.  The manuscript is certainly recommended for publication. 

There are minor corrections recommended that the authors can complete during the process of correction/proofreading. 

  • Point 1:It would be good if authors write a sentence at the end of the Introduction that describes the structure of the manuscript.

Reponse: We have added the structure of the manuscript in L109-114.

  • Point 2:Also references should be expend with ones which describe a similar problem in a different way and current state in this area.

Reponse: We have added the references of hailstorms, with the serial numbers from 11 to 14. Related revision is in L46. The details of hailstone references are depicted in L64-70, L93-99.

  • Point 3:Please expend and better explain the method i.e. the section with FFT.

Reponse: We have depicted the method of FFT in more detail in L131-132. The introduction of NCEP/NCAR reanalysis data , GDAS data and the calculation formulas of the whole layer water vapor flux have also been added in L142-157.

  • Point 4:Figures 5 and 7 need title of X-axis.

Reponse: The titles of X-axis has added in figure 6 and 8 (the original numbers of figures are 5 and 7).

  • Point 5:It would also be a good to read the text carefully  to correct English and typos.

Reponse: The English mistakes in grammar and typos have been corrected.

Reviewer 3 Report

The publication of the paper is useful for the operation of human-life and social activities. Before the publication a few re-considerations are necessary.

1)     Figure 2; Datum-line expressing the boundary between the land and sea is not clear. Please use more thick line.

2)     What is BJT?

3)     Figure 3; Please indicate the scale of the Vectors.

4)     Figure 6: The upper and lower figures are the same? If not same, why is the x-axis range is changed?

5)     In the paper, you revealed the relationship between the duration of hailstone and the air pressure gravity waves. However, the clear reason why the high air-pressure formats the hailstones is not clearly indicated. If possible, please add the explanation about the simple mechanism image of generation of hailstones.

Author Response

Response to Reviewer 3 Comments

Dear reviewer,

Thank you very much for your precious comments. According to your opinions, we have revised manuscript carefully. Hope it more suitable for publication. The revised part of article is highlighted in blue font. Moreover, due to the addition of references and figures, the serial number of references and figures in the most text have also modified.

The publication of the paper is useful for the operation of human-life and social activities. Before the publication a few re-considerations are necessary.

  • Point 1:Figure 2; Datum-line expressing the boundary between the land and sea is not clear. Please use more thick line.

Reponse: To better display the boundary between the land and sea, figure 3 (the toponomy of the surrounding and the region of study) is added to clearly show the boundary between the land and sea. The more thick line is also used in figure 2.

  • Point 2:What is BJT?

Reponse: The abbreviation of BeiJing Time is BJT. This is added in L102 when first quoted in the manuscript.

  • Point 3:Figure 3; Please indicate the scale of the Vectors.

Reponse: The scale of the whole layer water vapor flux has added in figure 4c (the original number of figure is 3).

  • Point 4:Figure 6: The upper and lower figures are the same? If not same, why is the x-axis range is changed?

Reponse: The figure 7a and figure7b (the original numbers of figures are 6a and 6b) both display the the frequency spectrum characteristics of gravity waves. To better analyze the frequency spectrum characteristics in more detail, the periods of gravity waves in figure 7a were less than 240 minutes, while the periods in figure 7b less than 80 minutes. The x-axis ranges of figure 7a and 7b have been revised to be same.

  • Point 5:In the paper, you revealed the relationship between the duration of hailstone and the air pressure gravity waves. However, the clear reason why the high air-pressure formats the hailstones is not clearly indicated. If possible, please add the explanation about the simple mechanism image of generation of hailstones.

Reponse: We add the section “4.4 The mechanism of generation of gravity waves”in L388-429, to explain the reason why the high pressure are closely related with hailstones. The abstract and conclusion of this manuscript also revised in L28-31, L457-464.

Round 2

Reviewer 3 Report

The text is well revised. Especially, Figure 3 is very useful for better understanding of the location of the target. The target region is Jilin Province?

If so, please refer to the province name at the former part. 

The explain at Figure 3 has a miss-spelling ( haistone >> hailstone)   

Author Response

Response to Reviewer 3 Comments

Dear reviewer,

Thank you very much for your precious comments. According to your opinions, we have revised manuscript carefully. Hope it more suitable for publication. The revised part of article is highlighted in blue font.

  • Point 1: The text is well revised. Especially, Figure 3 is very useful for better understanding of the location of the target. The target region is Jilin Province?

If so, please refer to the province name at the former part. 

Reponse: The region of study is Jilin Province. In figure 3a, the province name is depicted. Relevant description has been added in L179-180.

  • Point 2:The explain at Figure 3 has a miss-spelling ( haistone >> hailstone)   

Reponse: We have revised it.
